# Dissipation Behavior and Dietary Risk Assessment of Thiamethoxam, Pyraclostrobin, and Their Metabolites in Home-Style Pickled Cowpea

**DOI:** 10.3390/foods12183337

**Published:** 2023-09-06

**Authors:** Xumi Wang, Huanqi Wu, Kongtan Yang, Nan Fang, Hong Wen, Changpeng Zhang, Xiangyun Wang, Daodong Pan

**Affiliations:** 1College of Food and Pharmaceutical Sciences, Ningbo University, Ningbo 315211, China; wxm17815625975@163.com (X.W.); wuhuanqi2023@163.com (H.W.); 2State Key Laboratory for Managing Biotic and Chemical Threats to the Quality and Safety of Agro-Products, Ministry of Agriculture and Rural Affairs Key Laboratory for Pesticide Residue Detection, Institute of Agro-Products Safety and Nutrition, Zhejiang Academy of Agricultural Sciences, Hangzhou 310021, China; yangkongtan99@163.com (K.Y.); fn199198@hotmail.com (N.F.); dayiro@163.com (H.W.); zhangcp@zaas.ac.cn (C.Z.); 3College of Plant Protection, Jilin Agricultural University, Changchun 130118, China

**Keywords:** pickled cowpea, pesticide, metabolite, dissipation, degradation

## Abstract

In this study, the fate of two pesticides commonly used on cowpeas, thiamethoxam and pyraclostrobin, during the preparation of home-made pickled cowpeas was investigated using an improved QuEChERS method combined with UHPLC-MS/MS. Although pesticide residues were primarily distributed on cowpea samples, some were transferred to brine. The dissipation half-life of thiamethoxam on cowpea samples was significantly shorter than that of pyraclostrobin due to thiamethoxam’s higher water solubility. Thiamethoxam demonstrated a half-life of 5.12 ± 0.66 days, whereas pyraclostrobin exhibited a longer half-life of 71.46 ± 7.87 days. In addition, the degradation half-lives of these two pesticides in the whole system (cowpea and brine) were 45.01 ± 4.99 and 70.51 ± 5.91 days, respectively. This result indicates that the pickling did not effectively promote the degradation of thiamethoxam and pyraclostrobin. The metabolite clothianidin of thiamethoxam was not produced throughout the pickling process, but the metabolite BF 500-3 of pyraclostrobin was detected in cowpea samples. The detection rates for thiamethoxam, pyraclostrobin, and BF 500-3 in the 20 market samples were 10%, 70%, and 45%, respectively. However, the risk quotient analysis indicated that the risk of dietary intake of thiamethoxam and pyraclostrobin in pickled cowpeas by Chinese consumers was negligible.

## 1. Introduction

Cowpea (*Vigna unguiculata* L. Walp) is an economically important vegetable crop with high nutritional value and is grown widely in China [1]. Cowpeas are often processed into pickled cowpeas, which are popular among consumers because of their extended shelf life and unique flavor [2]. However, cowpea cultivation is vulnerable to insect pests and plant diseases. Therefore, pesticides are used continuously throughout the cowpea crop cycle to ensure good yields and minimize economic losses [3]. However, excessive use of pesticides will lead to severe pesticide residues on cowpeas and pose health risks to consumers. In 2010, a significant cowpea contamination incident occurred in Hainan, where pesticide residues exceeded their allowed limits [4]. In recent years, the problem of pesticide residues in cowpeas remains a challenge, prompting comprehensive management efforts nationwide. The presence of pesticide residues in pickled cowpeas made from fresh cowpeas is a matter of concern. Nonetheless, few studies have investigated pesticide residues in pickled cowpea. One study discovered carbendazim in 24 pickled cowpea samples, exhibiting the highest concentration at 43.8 μg·L^−1^ and the median concentration at 13.2 μg·L^−1^. Among the tested samples, 29.2% exceeded the maximum residue limit of carbendazim in cowpeas in China and 70.8% exceeded the limit in Australia [5].

Thiamethoxam (THI, CAS:153719-23-4) is a second-generation neonicotinoid insecticide that selectively inhibits the nicotinic acetylcholine esterase receptor in the nervous system of insects [6]. Clothianidin (CLO, CAS:210880-92-5) is another neonicotinoid insecticide and a THI metabolite [7]. Pyraclostrobin (PYR, CAS:175013-18-0) is a strobilurin fungicide that was discovered by BASF in 2000 [8]. PYR is widely used to manage fruits and vegetables, attributed to its ability to inhibit fungal cell mitochondrial respiration [9,10]. BF 500-3, the main metabolite of PYR in plants, bears a chemical structure similar to that of PYR, hence being regarded as equally toxic [11,12]. The chemical structures of THI, CLO, PYR, and BF 500-3 are shown in Figure 1. Previous studies have elucidated the detrimental effects of THI and CLO on the nervous and endocrine systems of mammals [13]. Moreover, PYR’s significant toxicity to non-target aquatic organisms, such as fish, invertebrates, and amphibians, was analyzed [14,15]. However, as registered pesticides on cowpeas, THI and PYR are widely used in cultivating cowpeas and inevitably cause severe contamination. For instance, a comprehensive study by Zhang et al. analyzed 1588 vegetable samples from organic vegetable bases, farmers’ markets, and supermarkets in Zhejiang, China, revealing a relatively high detection rate of THI in cowpea samples, reaching 27.4%, with the highest recorded residue level reaching 0.78 mg·Kg^−1^ [16]. Ma et al., in their study, found PYR to exhibit the highest detection rate among the 38 pesticides analyzed in various vegetables, including cowpeas [17]. Consequently, further research is necessary to examine the residue characteristics of THI, PYR, and their metabolites in fresh cowpeas and pickled cowpeas made from fresh cowpeas to ensure food safety.

Several residue studies have reported the presence of THI and PYR in cowpeas. In their study, Liao et al. investigated the degradation and metabolism of THI in cowpeas at different growth stages. Their group concluded that THI is safe during the sowing and seedling periods but presents significant risks during the peak production period [18]. Chen et al. studied THI’s dissipation, residue levels, and metabolites in cowpeas under field conditions. The results revealed a half-life of 0.8–1.6 days for THI in cowpeas, with the final residue levels of THI and its metabolite below the maximum residue levels established in the EU seven days before harvest [19]. Han et al. reported a rapid dissipation of PYR in cowpeas, characterized by half-lives of 1.5–2.3 days [20]. However, currently, there is a lack of relevant research on THI and PYR in pickled cowpea.

In this paper, the fate of THI, PYR, and their metabolites during cowpea pickling was studied. Including investigating the dissipation of THI and PYR in pickled cowpea samples, analyzing the changes in pesticide residues in the brine, studying the degradation of both pesticides during pickling, and monitoring their metabolites. In addition, the samples available on the market were analyzed to assess potential health risks for consumers.

## 2. Materials and Methods

### 2.1. Chemicals and Reagents

THI (99.65%, purity) and CLO (97.9%, purity) were manufactured by Dr. Ehrenstorfer GmbH (Augsburg, Germany) and were purchased from J&K Scientific Ltd. (Beijing, China). PYR (99.0%, purity) was purchased from Shanghai Pesticide Research Institute Co., Ltd. BF 500-3 (97.7%, purity) was purchased from Alta Scientific Co., Ltd. Acetonitrile and methanol of HPLC-grade were provided by Merck Co (Darmstadt, Germany). HPLC-grade formic acid (FA) was obtained from Anaqua Chemicals Supply (Wilmington, NC, USA) and ammonium acetate was purchased from Tedia Company (Fairfield, CA, USA). Analytical-grade acetonitrile was obtained from Shanghai Ling Feng Chemistry Reagent Co., Ltd. (Shanghai, China). Experimental water was obtained from Hangzhou QuChenShi Group Co., Ltd. (Hangzhou, China). Analytical-grade sodium chloride (NaCl) and anhydrous magnesium sulfate (MgSO_4_) were purchased from Sinopharm Chemical Reagent (Beijing, China). Sorbent PSA was purchased from Agilent Technologies (Tianjin, China). Pesticide stock solutions were prepared at 1000 mg·L^−^^1^ in HPLC-grade methanol and stored at 4 °C.

### 2.2. Materials

The fresh cowpeas and salt used in the experiment were purchased from local supermarkets, and the cowpeas were tested to be devoid of THI, CLO, PYR, and BF 500-3.

### 2.3. Pickling Process 

Experiments were carried out following the established home pickled cowpea processing method and the relevant literature for preparing the positive samples [21].

Each fermentation tank (Sichuan Shubo Co., Ltd., Chengdu, Sichuan, China) was filled with 200 g of clean and freshly cut cowpeas. Subsequently, 1 L of water containing 200 μL of Tween 80 was added to facilitate the dissolution of pesticides, and the resultant mixture was subjected to ultrasonic stirring for 3 min with the SK-8200H ultrasonic cleaner (Shanghai Kedao Ultrasonic Instrument Co., Ltd., Shanghai, China). For component analysis, 1 mL of the standard solution of THI (1000 mg·L^−1^) was added to half fermentation tanks, and 1 mL of the standard solution of PYR (1000 mg·L^−1^) was added to the remaining fermentation tanks. The liquid was subjected to an additional 3 min of ultrasound agitation and soaked for 1 h. Afterward, the solution was drained from the fermenter, and the cowpeas were rinsed with water to remove residual pesticides from the sample’s surface.

Cowpeas were air-dried and subsequently transferred to clean fermentation tanks. Subsequently, 500 mL of a 7% *w*/*v* salt solution was added dropwise to the fermentation tank, and the pickling procedure was conducted at 25 °C. Control samples, lacking the target pesticides, underwent identical treatment and were pickled under similar conditions. Three replicates were sampled from each treatment at different time intervals (0, 1, 2, 3, 5, 7, 10, 14, 28, and 42 days). For analysis purposes, the pickled cowpeas and brine used for pickling were collected. The cowpea samples were washed three times with water and then homogenized using a blender to minimize interference with the detection of pesticide residues in pickled cowpeas. The brine was filtered using filter paper and collected for analysis. These prepared samples were frozen at −20 °C until the extraction process.

### 2.4. Sample Pretreatment

The extraction of THI, PYR, and their metabolites from cowpea samples and brine was conducted with a modified QuEChERS (Quick, Easy, Cheap, Effective, Rugged, and Safe) approach [22]. Ten grams of the cowpea sample or 10 g of brine was accurately measured into a 50 mL polypropylene centrifuge tube. Next, 10 mL of acetonitrile was added as the solvent for extraction. The tube was mixed on a vortex mixer (Taizhou Nuomi Medical Technology Co., Ltd., Taizhou, Jiangsu, China) for 5 min, and 4 g of NaCl was added to facilitate the separation of acetonitrile from water. The mixture was centrifuged at 3040× *g* for 5 min using a low-speed centrifuge (Hunan Xiangyi Laboratory Instrument Development Co., Ltd., Changsha, Hunan, China), and the resulting acetonitrile solution was transferred into a clean test tube. Another 10 mL of acetonitrile was used for a repeated extraction procedure, and the obtained acetonitrile extracts were combined for further analysis. Based on the preliminary experimental results, it has been observed that repeated extraction enhances the recovery rate, particularly for THI, a pesticide with strong water solubility. Subsequently, a mixture of 1.6 mL was added to a 2 mL plastic tube containing 150 mg of magnesium sulfate and 50 mg of PSA. The plastic tube was vortexed for 30 s and then centrifuged at 4448× *g* for 5 min using a high-speed centrifuge (Hunan Xiangyi Laboratory Instrument Development Co., Ltd., Changsha, Hunan, China). The clear supernatant was filtered through a 0.22 μm membrane before performing UPLC-MS/MS (Waters Corp, Milford, MA, USA) analysis.

### 2.5. UPLC-MS/MS Analysis

The UPLC-MS/MS methodology exhibits exceptional sensitivity and can successfully detect trace amounts of pesticide residues. Its efficient separation capabilities enable simultaneous detection of multiple pesticides, while expertly avoiding the matrix effect. Moreover, the UPLC-MS/MS approach facilitates accurate identification and quantitative analysis of different pesticide variants, ensuring dependable and precise test results [23]. A column with reversed-phase properties (Acquity UPLC BEH C18, 2.1 × 100 mm, 1.7 μm, Waters Corp, Milford, MA, USA) was utilized for chromatographic separation. The temperature of the column and the flow rate were established at 40 °C and 0.2 mL·min^−1^, respectively. The mobile phase was acetonitrile (A) and 0.1% FA in water with 2 mM ammonium acetate (B). The isocratic mobile phase was utilized as follows: Mobile phase A (90%) and mobile phase B (10%) with an injection volume of 2 µL. 

Pesticide analysis was conducted in positive electrospray ionization (ESI+) mode on a triple-quadrupole mass spectrometer (XEVO TQ 125 MS, Waters, Milford, MA, USA). The capillary and extractor voltages were adjusted to 1.5 kV and 4 V, respectively. Source temperature and desolvation temperature were maintained at 115 °C and 450 °C, with a desolvation gas flow rate of 600 L·h^−1^. Detection was performed in multiple reaction monitoring mode (MRM). The quantification and confirmation of the MS/MS transition along with the optimized collision energy (CE) and retention time for target chemicals are listed in Appendix A.

### 2.6. Method Validation

The analytical parameters were evaluated following the European guideline SANTE/12682/2019 [24]. The developed method was validated by assessing the linearity, accuracy, precision, limit of quantification (LOQ), and matrix effect (ME) [25]. The linearity was determined by calculating the determination coefficient (R^2^) of the standard calibration curves for both matrix and solvent matching. The standard calibration curves of THI and CLO were generated at six concentration levels, including 0.01, 0.02, 0.05, 0.1, 0.2, and 0.5 mg·L^−1^, and the standard calibration curves of PYR and BF 500-3 were generated at six concentration levels, including 0.005, 0.01, 0.05, 0.1, 0.2, and 0.5 mg·L^−1^. The purpose of these evaluations was to ensure the accuracy and reliability of the analytical method. Recovery experiments with five replicates at three spiked levels were conducted for each matrix to assess the method’s accuracy. Precision was evaluated using the relative standard deviation (RSD, %). The method’s LOQ was determined as the minimum level at which effective fortification was demonstrated. The ME was determined by comparing the calibration curve slopes of the solvent and matrix [26].

### 2.7. Collection and Analysis of Real Samples

Twenty packs of pickled cowpeas were procured from online retailers and local markets. The cowpea samples were washed three times with water, ground using a grinder, and the brine collected from the bags. THI, PYR, and their metabolites were extracted following the procedure described in Section 2.4. 

### 2.8. Statistical Analysis 

The ME was calculated using Equation (1):(1)ME=SmSS
where *Sm* and *Ss* refer to the gradients of the calibration curves of the matrix and solvent, respectively.

The dissipation and degradation kinetics of THI and PYR in pickled cowpea and the whole system were determined using Equation (2), and the half-life (t_1/2_) was calculated using Equation (3) [27]:(2)$$C_{t}=C_{0}e^{-kt}\;$$
(3)$$t_{1/2}=ln2/k$$
where C_0_ is the initial residue concentration (mg·kg^−1^), C_t_ is the residue concentration (mg·kg^−1^) at time t (d) after application, and k is the rate constant of degradation.

The total residues of PYR were calculated using Equation (4) [28]:(4)$$C_{sum}=C_{PYR}+C_{BF\;500-3}\times \left(M_{PYR}/M_{BF\;500-3}\right)$$
where C_sum_ denotes the overall residue concentration of PYR; C_PYR_ and C_BF 500-3_ are the concentrations of PYR and BF 500-3, respectively; M_PYR_ and M_BF 500-3_ are the molecular weights of PYR (387.82) and BF 500-3 (357.80), respectively.

The potential human health risks associated with long-term consumption of pickled cowpea and the intake of THI and PYR were determined using applicable Equations (5) and (6) [29,30,31]:(5)$${RQ}_{d}=EDI/(ADI\times bw)$$
(6)$$EDI=\mathrm{Residue}\;\mathrm{of}\;\mathrm{pesticide}\left(mg\cdot {Kg}^{-1}\right)\times \mathrm{Intake}\;\mathrm{of}\;\mathrm{pickled}\;\mathrm{cowpea}\left(Kg\right)$$
where RQ_d_ is the dietary risk quotients, EDI is the estimated daily intake (mg), and ADI is the acceptable daily intake (mg·kg^−1^·bw). ADIs for THI, CLO, and PYR were 0.08, 0.1, and 0.03 mg·kg^−1^·bw, respectively [32]. Bw represents the mean weight of a specific age group within a population, and the average weight of adults in China is 63 kg [33]. Pickled cowpea’s recommended daily serving size is 0.0103 kg [32]. If RQ_d_ exceeds 1, there is an unacceptable risk associated with long-term dietary intake of pesticides. Conversely, if RQ_d_ is less than 1, the health risk is considered acceptable. During computation, any value below the LOQ is substituted with the LOQ [34].

## 3. Results and Discussion

### 3.1. Method Validation

The R^2^, MEs, RSDs, and average recoveries (ARs) are summarized in Table 1 and Table 2. THI, PYR, and their metabolites exhibited excellent linearity across the standard curve range, with R^2^ values exceeding 0.9942 in all instances. The ARs of THI, PYR, and their metabolites in pickled cowpea and brine ranged from 88% to 111%, with the RSDs ranging from 1.3% to 9.1%. These findings verified that the approach satisfied the validation parameters with recoveries between 70% and 120% and RSD ≤ 20%. The LOQs for THI and CLO in pickled cowpea and brine were 0.01 mg·kg^−1^. The LOQs for PYR and BF 500-3 in pickled cowpea and brine were 0.005 mg·kg^−1^. ME can affect the detection of pesticide residues, as different matrices can influence the ionization of the target compound, resulting in signal enhancement or attenuation [35]. ME values greater than 1 indicate signal enhancement, whereas ME values less than 1 indicate signal attenuation [25]. The findings suggest that matrix interference can be disregarded when the ME value falls between 0.8 and 1.2 [36]. Table 1 shows significant ME values, emphasizing the importance of employing matrix-matched calibrations to achieve more accurate results.

### 3.2. Dissipation and Degradation Behavior of Pesticides in the Pickled Process of Cowpea 

The target analytes were detected using UPLC-MS/MS in pickled cowpea samples and brine at different pickling times to analyze the fate of THI and PYR during pickling.

Figure 2A shows the residue changes of THI in pickled cowpea samples, with the concentration of THI declining rapidly, especially in the first 10 days, by nearly 70%. According to first-order kinetics, the kinetic equation for THI was Ct = 0.367e^−0.1353t^ with an R^2^ value of 0.6827, and the half-life of THI was determined to be 5.12 ± 0.66 days. Reusing brine during home cowpea pickling is a common practice to expedite the pickling time. However, this practice may result in the accumulation of pesticides. Therefore, it is essential to study the residues of pesticides in the brine. On the tenth day, the concentrations of THI in brine peaked at 0.1 mg·L^−1^, followed by a gradual decline, as shown in Figure 2B. The presence of THI in brine could be attributed to the continuous transfer of THI molecules from cowpea to brine during the pickling process. Additionally, the water solubility might influence the transfer of THI, and the amount transferred is limited. Upon consulting the literature, relevant studies were discovered. Dong et al. determined the transfer rate of nine different water-soluble pesticides, including THI and PYR, from dry tea to tea soup, and found that the transfer rate of pesticides with high solubility was much higher than that of pesticides with low solubility. Another more critical discovery is that the transfer rate of these pesticides began to decline after reaching the peak at a certain time, and with the delay of time, its transfer rate was lower than its degradation rate, which made the pesticide content in tea soup gradually decrease [37]. This phenomenon explains the rapid reduction of THI in pickled cowpeas during the early stages and poor curve fitting. More importantly, these results illustrate that pesticides are not a simple degradation behavior in the pickled cowpea samples, but a dissipation behavior. The degradation behavior was investigated in the whole system (cowpea and brine) to gain insights into the fate of THI in the pickling process of cowpea. Figure 2C illustrates the degradation curves of THI across the entire system, with a half-life of 45.01 ± 4.99 days. This result suggests that the degradation of THI was much slower during cowpea pickling than in the field, which has reported a half-life of 0.8 to 1.6 days in cowpea fields [19]. The reason for this result could be that the PH of cowpeas kept dropping during the pickling process, while there is information that THI is stable under acidic conditions [38]. CLO was not detected either in cowpea samples or the brine. 

Figure 2D illustrates the dissipation curves of PYR in pickled cowpea. Assuming first-order kinetics, the dissipation of PYR followed the equation Ct = 0.508e^−0.0097t^ exhibiting a reasonable fit with the R^2^ value of 0.8767. The half-life of PYR was 71.46 ± 7.87 days. The metabolite BF 500-3 of PYR was detected in pickled cowpeas on day 28, exhibiting an increasing concentration trend. Figure 2E shows that the concentration of PYR in brine remains relatively low with no significant change. The fluctuations in concentration might be attributed to systematic experimental errors. Compared to THI, PYR has lower water solubility (1.9 mg·L^−1^, 20 °C), suggesting minimal transfer from pickled cowpeas to the brine, and therefore PYR was slower to dissipate than THI. The degradation kinetic equation of the PYR is Ct = 0.107 e^−0.0098t^, with a half-life time of 70.51 ± 5.91 days, as shown in Figure 2F. However, PYR has a shorter half-life of 1.5–2.3 days in field experiments on cowpea [20], indicating that PYR degrades more slowly during the pickling process. According to the literature, PYR is a weakly acidic pesticide that exhibits considerable stability in acidic environments [39]. The slow degradation of THI and PYR during cowpea pickling may be related to their own properties, as carbendazim degradation during cowpea pickling has been reported to be relatively fast, with a half-life of approximately 14 days [5].

### 3.3. Real Sample Analyses

The study analyzes the levels of THI, PYR, and their metabolites in pickled cowpeas from various manufacturers to determine residue and transformation. Separate tests were conducted on cowpea samples and brine, with each sample tested three times. Table 3 indicates a 10% detection rate of THI in 20 samples, with a maximum concentration of 0.021 mg·kg^−1^, although its metabolite CLO was not detected. PYR was detected in 20 samples with a maximum concentration of 0.203 mg·kg^−1^ and a median concentration of 0.039 mg·kg^−1^, while its detection rate reached 70%. The metabolite BF 500-3 of PYR was detected in 45% of the samples, with a maximum concentration of 0.052 mg·kg^−1^. Notably, no target compounds were detected in the brine, which might be attributed to replacing brine during packaging. This was also beneficial to reduce the pesticide residue in pickled cowpea. These results provide evidence of the widespread use of THI and PYR in cowpeas and emphasize the need to investigate their residue behavior in pickled cowpea. 

### 3.4. Dietary Exposure Risk Assessment 

Dietary risk assessment is critical in evaluating potential health risks associated with consuming contaminated foods, considering both chronic and acute pesticide residue intake in the general population [40]. This study conducted a basic risk assessment using the residual data of actual samples. According to the 2014 report of JMPR [32], THI and CLO are considered separately because CLO is registered as a pesticide. In China, only the parent PYR is currently defined as a residue [32], but research in the United States and Canada suggests that the metabolite BF 500-3’s toxicity is comparable to that of PYR, necessitating the simultaneous consideration of both compounds in residue and dietary risk assessments [41]. PYR and its metabolite BF 500-3 were calculated simultaneously in this study to evaluate the risk more accurately. The maximum residue of THI, CLO, and PYR in 20 samples was evaluated as RQ_d_. Table 4 shows the RQ_d_ was much less than 1, which indicated the safety of the unevaluated samples as well. Therefore, the intake of THI and PYR through pickled cowpea is unlikely to pose a public health concern. Nevertheless, this study’s risk assessment outcomes have certain limitations and should be used solely as a reference. Moreover, other pesticides used on cowpea may also have residues in pickled cowpea, and their risk assessment should be considered.

## 4. Conclusions

The present study developed and validated a modified QuEChERS method coupled with UPLC-MS/MS to detect THI, PYR, and their metabolites in pickled cowpea and brine. The method demonstrated specificity and exhibited suitable linearity, recovery, and precision for analyzing these compounds in both matrices. The results indicated that the half-life of THI was 5.12 ± 0.66 days in pickled cowpeas, whereas PYR exhibited a longer half-life of 71.46 ± 7.87 days. The variability in dissipation rate could be attributed to the varying solubilities. THI, being highly soluble in water, readily migrated from pickled cowpeas to the brine during cowpea pickling, promoting its rapid dissipation. However, for the 42-day study period, the degradation of THI and PYR did not achieve the half-life in the system, suggesting the limited effectiveness of degradation during the pickling process. Analysis of the 20 cowpea samples revealed THI detection in 10% of the samples, while PYR and its metabolite BF 500-3 were detected in 70% and 45% of the samples, respectively. However, the dietary risk posed by these residues in the actual samples was deemed acceptable. This outcome validated the widespread use of THI and PYR in cultivating cowpea fields, highlighting the need to assess the associated risk posed by these two pesticides in pickled cowpea. In conclusion, these data emphasized the importance of controlling pesticide residues in pickled cowpeas and other pickled vegetables to ensure food safety. However, similar studies are still relatively few. In the future, we should carry out more relevant research and further explore methods of rapid degradation of these pesticides based on existing data to ensure the safety of these pickled vegetables.

## Figures and Tables

**Figure 1 foods-12-03337-f001:**
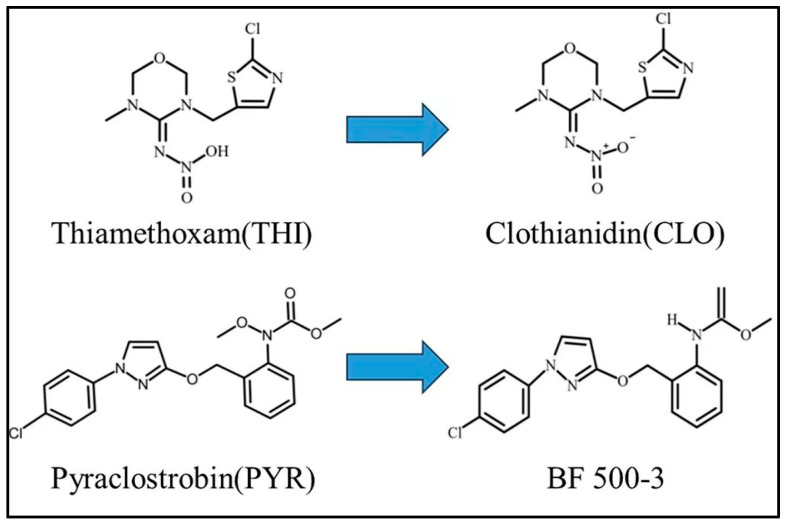
Chemical structures of thiamethoxam, clothianidin, pyraclostrobin, and BF 500-3.

**Figure 2 foods-12-03337-f002:**
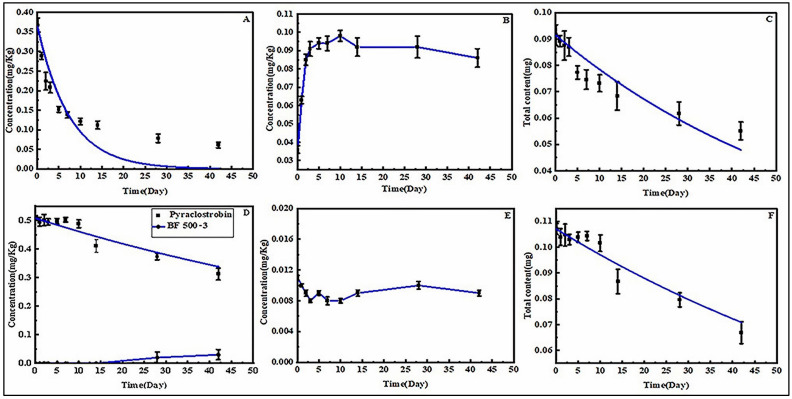
Dissipation, degradation kinetic curves, and variation curves of THI, PYR, and their metabolites. (**A**) Dissipation kinetic curves of THI in the pickled cowpea. (**B**) Variation curves of THI in brine. (**C**) Degradation kinetic curves of THI in the system. (**D**) Dissipation kinetic curves of PYR and variation curves of BF 500-3 in the pickled cowpea. (**E**) Variation curves of PYR in brine. (**F**) Degradation kinetic curves of PYR in the system.

**Table 1 foods-12-03337-t001:** The linear equations, correlation coefficient (R^2^), and matrix effects (MEs) of thiamethoxam (THI), pyraclostrobin (PYR), and their metabolites in pickled cowpea and brine matrices.

Pesticides	Matrices	Linear Range (mg·L^−1^)	Standard Curve Equations	R^2^	ME
THI	Solvent	0.01–0.5	y = 366887x − 2103	0.9988	/
	Cowpea		y = 34922x + 24	0.9987	0.10
	Brine		y = 55360x − 75	0.9965	0.15
CLO	Solvent	0.01–0.5	y = 55729x + 656	0.9978	/
	Cowpea		y = 7934.6x + 66	0.9980	0.14
	Brine		y = 13295x + 58	0.9988	0.24
PYR	Solvent	0.005–0.5	y = 746678x + 14518	0.9942	/
	Cowpea		y = 638872x + 5660	0.9960	0.86
	Brine		y = 653173x + 11039	0.9943	0.87
BF 500-3	Solvent	0.005–0.5	y = 2444053x − 14511	0.9987	/
	Cowpea		y = 1671010x − 2685	0.9997	0.68
	Brine		y = 1696270x − 6609	0.9978	0.69

**Table 2 foods-12-03337-t002:** The average recoveries (ARs), relative standard deviations (RSDs), and limits of quantification (LOQs) of THI, PYR, and their metabolites.

Matrices	Pesticides	Spiking Levels (mg·kg^−1^)	AR (%)	RSD (%)	LOQ (mg·kg^−1^)
Cowpea	THI	0.01	111	4.6	0.01
		0.1	106	9.1	
		1	105	2.7	
	CLO	0.01	95	6.0	0.01
		0.1	104	8.8	
		1	102	6.8	
	PYR	0.005	103	7.0	0.005
		0.1	103	1.3	
		1	100	5.8	
	BF 500-3	0.005	89	5.1	0.005
		0.1	108	2.0	
		1	103	5.0	
Brine	THI	0.01	102	6.9	0.01
		0.05	104	1.7	
		0.5	93	2.8	
	CLO	0.01	107	7.4	0.01
		0.05	91	2.0	
		0.5	101	5.3	
	PYR	0.005	88	8.3	0.005
		0.05	97	2.4	
		0.5	98	3.8	
	BF 500-3	0.005	104	5.5	0.005
		0.05	105	3.4	
		0.5	89	3.7	

**Table 3 foods-12-03337-t003:** Residue levels (mg·kg^−1^) of THI, CLO, PYR, BF 500-3, and total PYR (sum of PYR and BF 500-3, expressed as PYR) in different real samples.

Sample No.	THI	CLO	PYR	BF 500-3	Total PYR
1	0.021	<0.01	0.010	<0.005	0.015
2	0.015	<0.01	0.192	0.024	0.218
3	<0.01	<0.01	0.185	0.052	0.241
4	<0.01	<0.01	0.006	<0.005	0.011
5	<0.01	<0.01	<0.005	<0.005	<0.01
6	<0.01	<0.01	0.016	<0.005	0.021
7	<0.01	<0.01	0.066	0.009	0.076
8	<0.01	<0.01	0.028	0.009	0.038
9	<0.01	<0.01	<0.005	<0.005	<0.01
10	<0.01	<0.01	<0.005	<0.005	<0.01
11	<0.01	<0.01	0.047	0.01	0.058
12	<0.01	<0.01	<0.005	<0.005	<0.01
13	<0.01	<0.01	0.014	<0.005	0.019
14	<0.01	<0.01	0.034	0.008	0.043
15	<0.01	<0.01	0.045	0.005	0.050
16	<0.01	<0.01	0.203	0.049	0.256
17	<0.01	<0.01	0.014	<0.005	0.019
18	<0.01	<0.01	0.073	0.012	0.086
19	<0.01	<0.01	<0.005	<0.005	<0.01
20	<0.01	<0.01	<0.005	<0.005	<0.01

**Table 4 foods-12-03337-t004:** Dietary risk assessment of THI, CLO, and Total PYR in real pickled cowpea.

Pesticides	Maximum Residue (mg·Kg^−1^)	Dietary Risk Quotient		
		EDI	ADI	RQ
THI	0.021	0.0002	0.08	4 × 10^−5^
CLO	<0.01	0.0001	0.1	2 × 10^−5^
Total PYR	0.256	0.0026	0.03	1 × 10^−3^

## Data Availability

Data will be made available upon request.

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
