# Peer review of "Dissipation Behavior and Dietary Risk Assessment of Thiamethoxam, Pyraclostrobin, and Their Metabolites in Home-Style Pickled Cowpea"

_foods, 2023, doi:10.3390/foods12183337_

Round 1

Reviewer 1 Report

Dear Author, I reviewed the manuscript (foods-2567763) entitled Dissipation Behavior and Dietary Risk Assessment of Thiamethoxam, Pyraclostrobin, and Their Metabolites in Home-Style Pickled Cowpea. This manuscript presents relevant information about pesticide detection by chromatography. However, some sections of the submitted data can be improved. For this reason, I consider that this manuscript needs minor changes. 

Additional comments.

Highlight the advantages of using UPLC to identify pesticide compounds.

Check paragraphs extension in this manuscript.

Include an experimental design that contains statistical factors and variables of response in the statistical analyses applied to the findings of this research.

Include a statistical description in Figure 2. 

Compare the obtained findings with similar assays where similar pesticides were evaluated with UPLC techniques. 

Include future trends to keep working with the obtained data. 

Try to conclude with a general statement of the most relevant part of this study.

Author Response

We are very grateful to the reviewer for reviewing the paper so carefully, and we have tried our best to improve and make some changes to the manuscript.

Comment 1: Highlight the advantages of using UPLC to identify pesticide compounds.

Response: We followed your advice, presented the advantages of UPLC-MS/MS in the identification of pesticide compounds in Section 2.5 of the manuscript, and cited relevant literature to increase its persuasive power. (Lines 152-156)

Comment 2: Check paragraphs extension in this manuscript.

Response: We have followed your suggestion and made some modifications to the relevant questions.

Comment 3: Include an experimental design that contains statistical factors and variables of response in the statistical analyses applied to the findings of this research.

Response: We appreciate the good advice, but unfortunately, we can't change it. In the initial design of the experiment, we considered adding a control group of positive cowpeas to study the degradation of pesticides without curing, but fresh cowpeas soon decomposed and the experiment was stopped. In the final analysis, we can only refer to the degradation rate of the target pesticide in the field to measure the degradation rate of cowpea in the pickling process. This led to our experimental set of control factors and variables being fuzzy. We will follow your suggestion and carefully design the next experiment.

Comment 4: Include a statistical description in Figure 2.

Response: We appreciate the good advice. When analyzing the results of Figure 2, we only give a simple statistical description of several change curves. We believe that calculating the dissipation half-life and degradation half-life by fitting the dissipation curve and the degradation curve can better present our experimental results, so there is not much statistical description of it. We followed your advice and added descriptions to enrich the analysis of the results. (Lines 237-238)

Comment 5: Compare the obtained findings with similar assays where similar pesticides were evaluated with UPLC techniques.

Response: Following your advice, we added results from similar experiments to the results analysis section and compared them to enrich the manuscript. (Lines 279-281)

Comment 6: Include future trends to keep working with the obtained data.

Response: Following your advice, we added relevant content at the end of the manuscript.

Comment 7: Try to conclude with a general statement of the most relevant part of this study.

Response: We appreciate the good advice. We have also tried to state the most relevant parts of this study in a general way, but we have finally retained all relevant conclusions because there are relatively few similar studies, and we hope to provide more references for readers.

Thank you again for your valuable suggestions. Revisions will be indicated in blue in the new manuscript. I wish you a happy life.

Author Response

We are very grateful to the reviewer for reviewing the paper so carefully, and we have tried our best to improve and make some changes to the manuscript.

Comment 1: Suggest adding affiliation of Dr. Ehrenstorfer.

Response: Dr. Ehrenstorfer GmbH is a German company specializing in the production of pesticide standards, and we found that its description was consistent with our manuscript by looking at the literature, so it was not revised. But we have added information about the purchasing agency to the manuscript for the convenience of more readers. (Lines 92-93)

Comment 2: Suggest adding manufacturer description on fermenter used.

Response: We followed your advice and added information about the manufacturer of the fermenter to the manuscript. (Line110)

Comment 3: Suggest adding manufacturer description on ultrasonic stirring methods.

Response: We followed your advice and added information about the manufacturer of the ultrasonic cleaner to the manuscript. (Lines 113-114)

Comment 4: Suggest adding manufacturer description on centrifuge, the RPM unit does not describe gravity well since it varies depending on the size of rotor, suggest converting to g instead, or add rotor description.

Response: We appreciate it very much for this good suggestion, added information about the manufacturer of the centrifuge to the manuscript, and We also converted RPM to g. (Line 138, 147)

Comment 5: All equations not displaying correctly in PDF.

 Response: Unfortunately, we did not find any problems with the formula in the PDF manuscript, so we did not make any changes. We guess that it may be due to errors caused by software incompatibility. We sincerely hope that you will give us another chance if there really is a mistake.

Comment 6: Read samples (n=3), but table and relating discussion showed 20 samples.

Response: The statement "n = 3" signifies that the testing process involves dividing each sample into three parallel samples to enhance the accuracy and reliability of the test results. Although it was initially included in the manuscript, we have since deleted this information to avoid any confusion for the reader. (Line 302)

Comment 7: Formatting

Response: We have corrected the wrong format. (Line 322)

Comment 8: What does QuEChERS stands for? No discussion of QuEChERS found before conclusion.

Response: Quenchers is a kind of rapid sample pretreatment technology used in the detection of agricultural products. Due to our carelessness, which resulted in a lack of a clear statement in the manuscript, we added relevant descriptions in the sample pretreatment section of the new manuscript and cited the literature to enable the reader to better understand it. (Lines 132-134)

Thank you again for your valuable suggestions. Revisions will be indicated in yellow in the new manuscript. I wish you a happy life.

Reviewer 3 Report

My compliments to the authors. The experimental project has been set up in a valid way and the results obtained are very interesting. We must also take into account the fact that it is NEVER easy to perform chemical analyzes to search for similar residues in food.
The article was written with correctness and diligence. I have NOT noticed particular errors of concept and writing, but I have highlighted two small typos in the text of my review which I enclose. I ask the Authors to take note of them and to correct them.

The article was written in correct English, even the scientific aspects were clearly stated in English. I have no specific remarks to make.

Author Response

We are very grateful to the reviewer for reviewing this paper so carefully, and we are very pleased that our manuscript has received your approval. Revisions will be indicated in green in the new manuscript. I wish you a happy life.
